# SkipVSR: Adaptive Patch Routing for Video Super-Resolution with Inter-Frame Mask

## ABSTRACT

Deep neural networks have revealed enormous potential in video super-resolution (VSR), yet the expensive computational expense limits their deployment on resource-limited devices and actual scenarios, especially for restoring multiple frames simultaneously. Existing VSR models contain considerable redundant filter, which drag down the inference efficiency. To accelerate the inference of VSR models, we propose a scalable method based on adaptive patch routing to achieve more practical speedup. Specifically, we design a confidence estimator to predict the aggregation performance of each block for adjacent patch information, which learns to dynamically perform block skipping, i.e., choose which basic blocks of a VSR network to execute during inference so as to reduce total computation to the maximum extent without degrading reconstruction accuracy dramatically. However, we observe that skipping error would be amplified as the hidden states propagate along with recurrent networks. To alleviate the issue, we design Temporal feature distillation to guarantee the performance. This proposal essentially proposes an adaptive routing scheme for each patch. Extensive experiments demonstrate that our method can not only accelerate inference but also provide strong quantitative and qualitative results with the learned strategies. Built upon an BasicVSR model, our method achieves a speedup of 20% on average, going as high as 50% for some images, while even maintaining competitive performance on REDS4.

## KEYWORDS

Adaptive Inference, Video Super-Resolution, Efficient Methods

## 1 INTRODUCTION

Video Super-Resolution (VSR) aims to reconstruct a high-resolution video from its corresponding low-resolution counterpart by filling in missing details, which has been widely used in remote sensing, satellite imagery, surveillance and security. As the development of Deep Neural Networks (DNNs), plenty of DNN-based methods are proposed for VSR. Although existing CNN-based video super-resolution works have been made promising performance over the past few years, the video super-resolution in real application for smartphones and TV monitors need huge computation cost and unable to respond in real time. That is because recovering video always requires the integration of multi-frames, which usually come at a high computational cost and high memory footprint. That means, it is necessary to design an accelerate strategy for video super-resolution networks to be more applicable.

More important, most of the deep learning models perform inference in a static manner. Both the computational graph and the network parameters are fixed once trained, which may limit their representation power, efficiency and interpretability. Therefore, it's an issue that deserves to be explored.

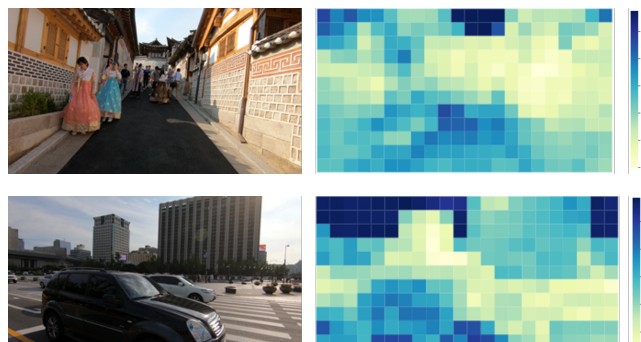

**Figure 1: The heatmap about patch PSNR index of two frames in REDS dataset. The PSNR is calculated between low-resolution image and high resolution image super-resolved by pre-trained BasicVSR [2] model. A darker color indicates a higher PSNR value, suggesting better results and easier recovery. On the other hand, lighter colors indicate lower PSNR values, implying greater difficulty in recovering the image. It shows that images with complex content or intricate textures tend to have lighter colors in the heat map, indicating a higher level of difficulty in achieving satisfactory recovery results. Conversely, images with simpler content or smoother textures exhibit darker colors, indicating a lower level of difficulty in the recovery process.**

To alleviate above issue, we introduce a new direction for effective and efficient VSR. To accelerate the inference process and train once for all scenes with different resource constraints, we design an adaptive patch routing scheme for the VSR task for the first time.

Currently, VSR methods can be mainly classified into two categories: parallel models and recurrent models. The parallel methods simultaneously enhance the recovery of all frames, and the recurrent model incorporates temporal dependencies by considering video features in the sequential manner, which means that the restoration for each frame depends on the information obtained from previous frames. Both proposals have their advantages and trade-offs, recurrent networks are widely used in VSR models to extract temporal features, consisting of residual blocks (e.g., BasicVSR[2] has 60 residual blocks). Therefore, we focus on the acceleration strategy for recurrent model and choose classical model BasicVSR [2] as our backbone.

In the field of single image super-resolution, there have been advancements in adapting subnetwork to accommodate varying computational demands in different regions. For instance, ClassSR [15] utilizes a classification scheme based on PSNR index to assign patch

regions to pretained models with different sizes; APE [23] introduces an adaptive patch exiting strategy for single image super-resolution, which estimates. while dynamic networks have been explored in single image super-resolution tasks, the considerations in VSR are different due to the need to incorporate temporal and spatial information from multiple frames.

To implement this idea, we need to address two challenges: 1) how to automatically find the effective routing for each image patches in video under different consumption Settings; and 2) how to effectively solve the significant drop in performance caused by error accumulation.

To explore the optimal path in our proposal, we introduce a decision mechanism after each residual block to determine whether next residual block shold be exexuted or skipped. For precise prediction, we develop a confidence estimator to give an guidance for execution. During training, we utilize intermediate feature difference as surpervision to constrain the training; while inference, we can obtain the optimal effective sub-network by adjusting the threshold. By this way, we can achieve once-for-all model. Furthermore, we have observed that when performing block skipping, there is an inevitable introduction of errors in the feature dimensions. In contrast to single image super-resolution, VSR tasks involve the fusion of features from previous frames. Thus, these error will accumulate layer by layer, which can lead to the model collapse. Therefore, we adopt temporal feature alignment, which involves leveraging a pre-trained model to correct the fratures at specific layers and maintain the consistency of features across frames.

In summary, our contributions can be concluded as :

(1) In this paper, we proposed a novel framework based on BasicVSR to accelerate VSR networks. To the best of our knowledge, we are one of the first to design a adaptive acceleration strategy for VSR.

(2) The patch-specific subnetwork is guided by the decision of the confidence estimator. Meanwhile, a GT constraint is also introduced to accurate prediction.

(3) Extensive experimental results demonstrate that our proposal can obtain great performance and efficiency trade-off, while achieving once-for-all scalable model.

## 2 RELATED WORK

### 2.1 Video Super-resolution

VSR models can leverage additional information from neighboring LR frames for restoration, as demonstrated in previous works such as [3, 5, 9, 10, 17, 26, 27, 29, 31, 32]. These methods exploit different techniques for alignment and fusion of LR frames.

Earlier VSR methods, such as [1, 21, 27], estimate the optical flow between LR frames and perform spatial warping for alignment. Then, more recent proposals have adopted iplicit alignment strategies. Instead of image-level motion alignment, TDAN [22] and EDVR [24] operate alignment at the feature level, aligning features from different frames by deformable convolutional [4] layers. EDVR further improves upon TDAN by introducing coarse-to-fine deformable alignment and a new spatial-temporal attention fusion module. RSDN [9] utilizes a recurrent detail-structural block and a hidden state adaptation module to mitigate the impact of appearance changes and error accumulation. Recently, BasicVSR [2]

discovered that bidirectional propagation coupled with a simple optical flow-based feature alignment can enhance performance. Likewise, in the work by Yi et al. [28], a bidirectional propagation framework is employed to exploit LR frames and estimate hidden states from the past, present, and future frames. To address the issue of huge consumption, Xiao et al. [25] proposed a space-time knowledge distillation scheme for VSR model compression. However, these VSR methods often require high computational costs, impeding their application on resource-limited devices. In contrast, our work focuses on designing an adpative patch routing strategy to expedite the inference of VSR models. By acceleratig the patch routing process, we aim to reduce computational requirements while maintaining restoration performance.

### 2.2 Content-Aware DNN

Deep neural networks(DNNs) are playing an important role in video super-resolution area, as shown in section 2.1. However, most of the deep learning models perform inference in a static manner, i.e., both the computational graph and the network parameters are fixed once trained, which may limit their representation power, efficiency and interpretability. Dynamic networks [7], as opposed to static ones, can adapt their structures or parameters to the input during inference, and therefore enjoy favorable properties that are absent in static models.

Motivate by the fact that different regions may have diverse computational demands, there are some methods about dynamic network in single image super-resolution. ClassSR [15] classifies the patch region as three kinds according to the PSNR of each path, and then deploy pretrained models with different sizes for restoring them. This method requires evaluating and counting the PSNR value of the whole dataset and manually classifying them, which is inconvenient. Then APE [23] proposes an adaptive patch exiting strategy for single image super-resolution, which estimates the exiting layer directly based on the degree of path recovery. These methods have greatly achieved performance and efficiency trade-off. However, the temporal and spatial information between multi frames must be considered in video super-resolution task.Since we cannot directly apply a general dynamic inference strategy for VSR, we explore the properties of VSR networks and develop an adaptive patch skipping scheme in this paper.

## 3 METHOD

### 3.1 BasicVSR

The BasicVSR [2] model is currently considered the state-of-the-art model, and it is chosen as the backbone network for the approach in this chapter. Now, let's provide a brief introduction to BasicVSR [2].

$$
\begin{aligned}
h_i^b &= F_b(x_i, x_{i+1}, h_{i+1}^b) \\
h_i^f &= F_f(x_i, x_{i-1}, h_{i-1}^b)
\end{aligned}
\tag{1}
$$

As shown in Figure 2, BasicVSR [2] is a classic bidirectional recurrent network composed of three modules: the backward feature fusion module (represented by the red module $F_b$ in Figure 2(a)), the forward feature fusion module (represented by the blue module $F_f$ in Figure 2(a)), and the upsampling module (represented by the yellow module $U$ in Figure 2(a)). Given a low-resolution image $x_t$

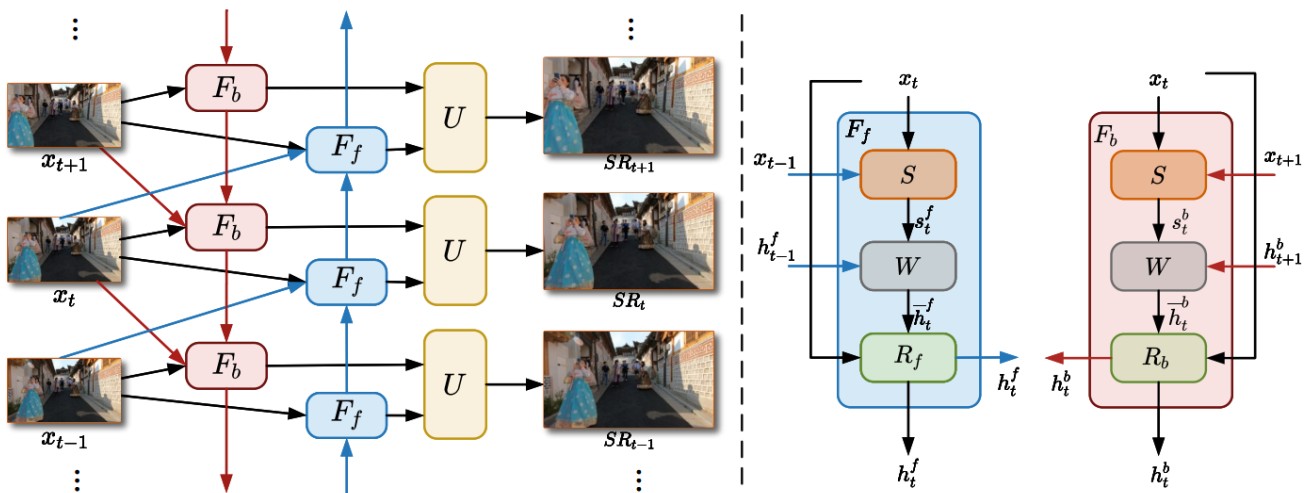

(a) The overview of BasicVSR    (b) The structure of forward/backward inference

Figure 2: The overview of BasicVSR

at the $t$-th frame, the forward feature fusion module $F_f$ extracts and fuses features by referring to the previous frame $X_{t-1}$, while the backward feature fusion module $F_b$ processes the image by referring to the next frame $X_{t+1}$. In other words, these two modules recursively fuse information from both ends of the video sequence.

The processing of these modules can be represented by the equations in (1). Finally, the upsampling module $U$ combines the outputs of $F_b$ and $F_f$ to reconstruct the super-resolution image $SR_t$ corresponding to the current frame.

As shown in Figure 2(b), both $F_b$ and $F_f$ modules have the same structure, consisting of motion estimation $S$, spatial transformation $W$, and residual blocks $R_f/R_b$. In this chapter, the redundancy in the residual blocks $R_f/R_b$ of the forward and backward recurrent feature fusion modules is explored and analyzed. By dynamically adjusting the inference process and reducing unnecessary computations, the efficiency of the model is improved while maintaining image quality.

## 3.2 Overview

For a video sequence $X_T \in \mathbb{R}^{T \times H \times W \times 3}$ consisting of T frames of low-resolution images, the corresponding high-resolution video image sequence $SR_T \in \mathbb{R}^{T \times sH \times sW \times 3}$ is obtained through model reconstruction, where s represents the image upscaling factor, and H, W, and 3 represent the height, width, and number of channels of the input frames, respectively.

In the forward feature fusion module, we take the dynamic inference of the residual block group $R$ as an example to explain. The residual block group $R$ performs feature fusion between adjacent frames in the order from the first frame to the last frame.

$$h^t = R(x_i, \overline{h}^t) \qquad (2)$$

For the $t$-th frame, the input to the residual block group $R$ is the current frame image $x_i$ and the spatially compensated feature $\overline{h}^t$ of the reference frame, and the output is the fused image feature $h^t$.

This can be expressed as Equation (2). The module $R$ consists of M residual blocks, and the fusion operates sequentially on the input features, resulting in the output features $h_m^t$ of the m-th fusion module, with the input features $h_{m-1}^t$.

$$h_m^t = R_m(h_{m-1}^t) \qquad (3)$$

In a video sequence, adjacent frames exhibit high similarity, known as inter-frame redundancy. Performing complete multi-layer fusion operations on all input features in the fusion module may lead to resource wastage. Therefore, this chapter proposes an adaptive mask prediction module that generates a binary mask $M_m^t$ (where m denotes the m-th residual block in R and t denotes the t-th frame) based on the similarity between the input frame and the reference frame features. This mask predicts whether to perform full fusion operations or partial fusion operations to reduce computation. The process is as follows:

$$\overline{h}_m^t = M_m^t \odot h_m^t + (1 - M_m^t) \odot \overline{h}_{m-1}^t \qquad (4)$$

Here, $\odot$ denotes element-wise multiplication.

In addition, in order to mitigate the issue of error accumulation when recursively propagating and fusing features of adjacent frames in BasicVSR [2], this chapter maps features at specific layers to a high-dimensional space and imposes constraints during training to ensure the accuracy of high-frequency details. For detailed explanation, please refer to Section 3.4.

## 3.3 Adaptive Mask Prediction Module

This section provides a detailed explanation of the computation process for generating the adaptive mask $\boldsymbol{M}_m^t$ based on inter-frame redundancy.

The inter-frame redundancy refers to the similarity between the input features received by the current fusion module. In this chapter, the difference between the current frame feature $\overline{h}_m^t$ and the

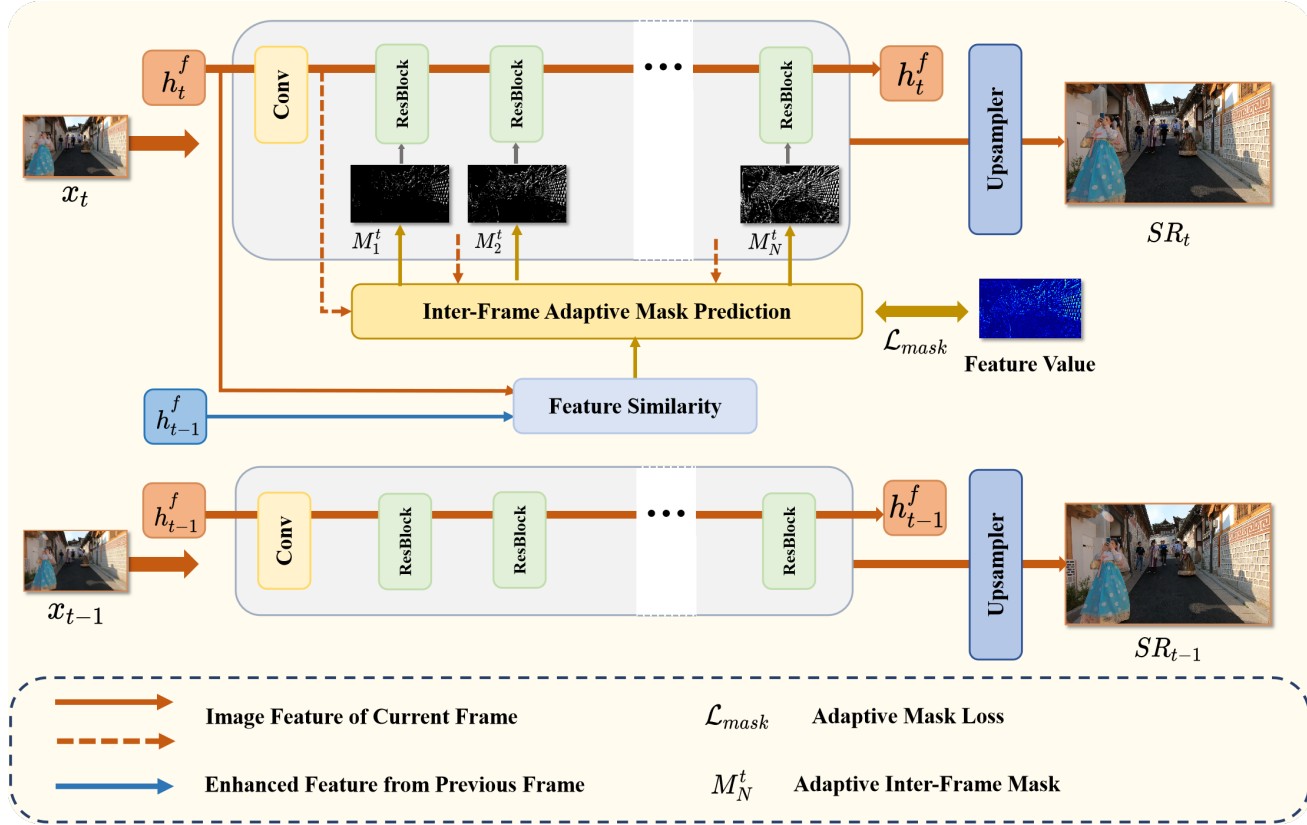

**Figure 3: (a). The basic architecture of the VSR methods with the bidirectional recurrent network. The forward and backward networks both consists of numerous residual blocks. The extracted feature from multi-frame images will input into the upsampler, which will restore videos. (b). The adaptive patch routing strategy proposed by this paper. Given a new video, multi-frame images will input into forward network and backward network to generate temporal and spatial information. We design a confidence estimator to evaluate the importance of each residual blocks. If the confidence value is below a given threshold, the patches will be not excuted by next residual block. Finally, the SR patches are merged to the output image.**

reference frame feature $\overline{h}_m^{t-1}$ is used as the inter-frame redundancy $\triangle T$:

$$\triangle T = ||\overline{h}_m^t - \overline{h}_m^{t-1}|| \tag{5}$$

A small adaptive mask generator is employed, which takes the inter-frame redundancy $\triangle T$ as input and generates the corresponding fusion confidence score $\hat{S}_i$:

$$\hat{S}_i = \sigma(W * g(h_m^{\triangle T}) + b) \tag{6}$$

The generator consists of a fully connected layer and a global average pooling layer. Here, $g$ represents the global average pooling operation, and $W$ and $b$ are the weights and biases of the fully connected layer. The same generator is shared across all layers (separate generators are used for the forward and backward fusion modules), and it is jointly trained with the backbone network.

As the modules deepen, the fusion effect on image patches reaches a bottleneck, and the feature maps exhibit minimal changes. To ensure the correctness of the adaptive mask generation, the explicit variation $S_i$ between the current layer feature $h_{m,t}^S$ of the

dynamic network $S$ and the target layer feature $h_{target,t}^T$ of the original network $T$ is used as supervision for the generator:

$$S_i = 1 - \theta||h_{m,t}^S - h_{target,t}^T|| \tag{7}$$

where $\theta$ represents the Tanh activation function, and a target layer is selected every 6 layers. During the training phase, the confidence score $S_i$ is used to constrain the generator by applying an $L_2$ loss between the generated fusion confidence $\hat{S}_i$ and the supervised variation $S_i$:

$$L_i = ||\hat{S}_i - S_i||_2^2 \tag{8}$$

During the inference phase, the method uses the obtained fusion confidence scores to generate a binary mask by applying a threshold:

$$M_{m,n}^t = \begin{cases} 1, & \text{if } S_i > \alpha \\ 0, & \text{otherwise} \end{cases} \tag{9}$$

The calculated binary mask $M_{m,n}^t$ is then used in Equation (4) for further computation.

## 3.4 Temporal Feature Alignment

Based on the structure design of BasicVSR [2], the restoration of the t-th frame image is achieved based on the (t-1)-th frame and the (t+1)-th frame. If there is a small deviation in generating the hidden layer features $h_f$ and $h_b$ in the dynamic network, this deviation will be propagated frame by frame and amplified during the network inference, leading to model collapse and inability to reconstruct subsequent frames.

To address this issue, this chapter aligns specific high-dimensional features of the dynamic network $S$ with the original model $T$ to correct the errors and maintain model accuracy.

Inspired by spatial attention map distillation, this chapter proposes to generate high-dimensional features through spatial feature mapping $\mathcal{F}$. Here, $H_t^T \in \mathbb{R}^{C \times W \times H}$ and $H_t^S \in \mathbb{R}^{C \times W \times H}$ represent the feature maps of the t-th frame image in the original network and the dynamic network, respectively. Here, $C$, $H$, and $W$ represent the number of channels, height, and width of the feature maps, respectively. The mapping function $\mathcal{F}$ can be defined as one of the following three operations:

$$
\begin{aligned}
\mathcal{F}_{\text{sum}}(H_t) &= \sum_{i=1}^{C} |H_{t,i}| \\
\mathcal{F}_{\text{sum}}^2(H_t) &= \sum_{i=1}^{C} |H_{t,i}|^2 \\
\mathcal{F}_{\text{max}}^2(H_t) &= \max_{i=1}^{C} |H_{t,i}|^2
\end{aligned}
\tag{10}
$$

where $H_{t,i}$ represents the i-th slice in the feature channel dimension. $\mathcal{F}_{\text{sum}}^2(H_t)$ is used as the mapping function, which gives more weight to high-frequency details and describes scene details more clearly and accurately through global computation.

During the training of the dynamic network, specific high-dimensional feature maps $Q_t^S$ and $Q_t^T$ are selected from a certain interval of layers in the dynamic network and the original network, and their differences are constrained by the $L_2$ loss to ensure consistency between the dynamic network features and the original network features, preserving high-frequency details. The error correction constraint is as follows:

$$
L_{\text{error}} = \frac{1}{T} \sum_{t=1}^{T} ||\mathcal{F}_{\text{sum}}^2(H_t^S) - \mathcal{F}_{\text{sum}}^2(H_t^T)||_2^2
\tag{11}
$$

Here, $L_{\text{error}}$ represents the $L_2$ norm distance, and $T$ is the number of frames in the input sequence.

## 3.5 Training Process

According to previous methods EDVR [? ] and BasicVSR [? ], this chapter employs the Charbonnier loss as the reconstruction loss to constrain the training of the backbone network:

$$
L_{\text{rec}} = \sqrt{||SR_t - HR_t||^2 + \epsilon^2}
\tag{12}
$$

Here, $\epsilon$ is set to $1e^{-6}$. $SR_t$ and $HR_t$ denote the high-resolution image reconstructed for the t-th frame and the corresponding ground truth high-resolution image, respectively. Based on this design, the overall training loss function of the dynamic network includes the reconstruction loss, the adaptive mask generation loss, and the error correction constraint, as follows:

$$
L = L_{\text{rec}} + \lambda_1 L_{\text{mask}} + \lambda_2 L_{\text{error}}
\tag{13}
$$

where $\lambda_1$ and $\lambda_2$ are weight coefficients, and $L_{\text{mask}}$ and $L_{\text{error}}$ represent the constraints for adaptive mask training and error correction, respectively.

## 4 EXPERIMENTS

### 4.1 Experiments Settings

We adopts widely-used datasets for training: REDS[19] and Vimeo-90K[27]. For REDS, following, we use the REDS4 dataset as our test set. We additionally define REDSval4 as our validation set. The remaining clips are used for training. We use Vid4[16], UDM10[29], and Vimeo-90K-T[27] as test sets along with Vimeo-90K[27]. We train and test our models with 4× downsampling using two degradations - Bicubic (BI) and Blur Downsampling (BD) as BasicVSR did. For BI, the MATLAB function "imresize" is used for downsampling. For BD, we blur the HR images by a Guassian filter with $\theta = 1.6$, followed by a subsampling every four pixels.

We use pre-trained SpyNet[20] as our flow estimation, and fix the parameters of the flow estimator are fixed. The patch size of input LR frames[18] is 48 × 48. We adopt Adam[14] optimizer, $\theta = 10^{-8}$, Cosine Annealing scheme and we use $L_1$ loss as loss function. We pretrain BasicVSR as VSR network backbone. Experiments are conducted on a server with Pytorch 1.10 and V100 GPUs.

The batch size is 16 and the LR patch size is 48. We use Adam optimizer, where $\beta_1$ is set to 0.9 and $\beta_2$ is set to 0.999. During testing, we first split LR images into patches of size 48 with stride 46 unless otherwise specified. Then the LR patches are super-resolved in parallel, and the parallel size can be tuned to fit the computational resource. Finally, the SR patches are merged to obtain the complete SR images by weighting overlapping areas. The Peak Signal-to-Noise Ratio (PSNR) and Structural Similarity (SSIM) calculated on RGB channels are adopted as the evaluation metrics to measure SR performance. We use FLOPs to evaluate the computational cost and the practical running time is benchmarked on NVIDIA V100 GPUs.

### 4.2 Evaluation of SkipVSR

**Performance Results.** To evaluate the effectiveness and applicability of our proposed method, we conduct experiments by applying it to the classical method BasicSR [2] and compare iy with state-of-the-art and representative VSR netwroks. For a fair comparison, we not only adopt the model trained with our adaptive skipping strategy, but also compare models with varing scale sizes. Specifically, we set different confidence thresholds during the inference process.

We evaluate the performance of SkipVSR on various datasets, as shown in Table 1. It shows that VSR networks with SkipVSR can achieve comparable performance compared to original SR networks in terms of PSNR and SSIM. This comparison validates that our adaptive patch routing can provide insight into the diminishing return and role of individual layers.

**Efficiency Results** As for the efficiency of SkipVSR, Table 2 shows the performance under different computational costs as BasicVSR [2]. Although our method add lightweight decisive network, it only consists of several MLP layers and its FLOPs are negligible. It illustrated that our method can reduce the computational cost of

Table 1: Quantitative comparisons of state-of-the-art methods and SkipVSR. To compare the performance of SkipVSR under the same computation constraints, the confidence threshold of SkipVSR is set 0. Parameters, FLOPs, PSNR and SSIM on various representative datasets with scaling factors ×4 are reported in the table. The changes in PSNR and FLOPs indexes corresponding to BasicVSR and SkipSR are marked in red and blue, respectively. Note that all the efficiency indexes (Params and FLOPs) are measured under the setting of LR images as $180 \times 320$ resolution on all scales.

| Methods | Params (M) | FLOPs (G) | Runtime (ms) | BI degradatioin | | | BD degradatioin | | |
| | | | | REDS4 [19] | Vimeo-90K-T [27] | Vid4 [16] | UDM10 [29] | Vimeo-90K-T [27] | Vid4 [16] |
|---|---|---|---|---|---|---|---|---|---|
| Bicubic | - | - | - | 26.14/0.7292 | 31.32/0.8684 | 23.78/0.6347 | 28.47/0.8253 | 31.30/0.8687 | 21.80/0.5246 |
| VESPCN [1] | - | - | - | - | - | 25.35/0.7557 | - | - | - |
| SPMC [21] | - | - | - | - | - | 25.88/0.7752 | - | - | - |
| TOFlow [27] | 1.4 | 274.9 | 1610 | 27.98/0.7990 | 33.08/0.9054 | 25.89/0.7651 | 36.26/0.9438 | 34.62/0.9212 | - |
| DUF [13] | 5.8 | 1645.8 | 974 | 28.63/0.8251 | - | - | 38.48/0.9605 | 36.87/0.9447 | 27.38/0.8329 |
| RBPN [8] | 12.2 | 8516.0 | 1507 | 30.09/0.8590 | 37.07/0.9435 | 27.12/0.8180 | 38.66/0.9596 | 37.20/0.9458 | - |
| EDVR-M [24] | 3.3 | 304.2 | 118 | 30.53/0.8699 | 37.09/0.9446 | 27.10/0.8186 | 39.40/0.9663 | 37.33/0.9484 | 27.45/0.8406 |
| PFNL [30] | 3.0 | 940.0 | 295 | 29.63/0.8502 | 36.14/0.9363 | 26.73/0.8029 | 38.74/0.9627 | - | 27.16/0.8355 |
| TGA [11] | 5.8 | 694.1 | 236 | - | - | - | - | 37.59/0.9516 | 27.63/0.8423 |
| RLSP [6] | 4.2 | 82.3 | 49 | - | - | - | 38.48/0.9606 | 36.49/0.9403 | 27.48/0.8388 |
| RSDN [9] | 6.2 | 355.7 | 94 | - | - | - | 39.35/0.9653 | 37.23/0.9471 | 27.92/0.8505 |
| RRN [12] | 3.4 | 108.7 | 45 | - | - | - | 38.96/0.9644 | - | 27.69/0.8488 |
| FastDVDnet* [25] | 2.6 | 64.3 | - | - | 36.12/0.9348 | 26.14/0.8029 | - | - | - |
| BasicVSR [2] | 4.9 | 338.5 | 57 | 30.65/0.8735 | 36.43/0.9372 | 26.54/0.7923 | 39.12/0.9650 | 36.75/0.9400 | 27.03/0.8444 |
| SkipVSR-BasicVSR | 4.9 | 338.5 | 58 | 30.60/0.8726 | 36.39/0.9365 | 26.54/0.7924 | 39.05/0.9645 | 36.73/0.9398 | 26.98/0.8434 |

Table 2: Efficiency evaluation of BasicVSR and SkipVSR with different threshold. In order to satisfy various scenarios with different upper limits of computational resources, we design SkipVSR, which can be flexibly varied by adjusting the confidence threshold after training whole model once. Parameters, FLOPs, PSNR and SSIM on various representative datasets with scaling factors ×4 are reported in the table. The changes in FLOPs and runtime indexes are marked in red and blue, respectively. Note that all the efficiency indexes (Params, FLOPs and Runtime) are measured under the setting of LR images as $180 \times 320$ resolution on all scales.

| Methods | Threshold | FLOPs (G) | Runtime (ms) | BI degradatioin | | | BD degradatioin | | |
| | | | | REDS4 [19] | Vimeo-90K-T [27] | Vid4 [16] | UDM10 [29] | Vimeo-90K-T [27] | Vid4 [16] |
|---|---|---|---|---|---|---|---|---|---|
| BasicVSR [2] | − | 338.5 | 57 | 30.65/0.8735 | 36.43/0.9372 | 26.54/0.7923 | 39.12/0.9650 | 36.75/0.9400 | 27.03/0.8444 |
| SkipVSR-BasicVSR | 0 | 338.5(100%) | 58 | 30.60/0.8726 | 36.39/0.9365 | 26.54/0.7924 | 39.05/0.9645 | 36.73/0.9398 | 26.98/0.8414 |
| SkipVSR-BasicVSR | 0.9 | 324.9(96%) | 57 | 30.41/0.8700 | 36.25/0.9342 | 26.39/0.7910 | 38.89/0.9634 | 36.60/0.9378 | 26.78/0.8212 |
| SkipVSR-BasicVSR | 0.92 | 297.8(88%) | 55 | 30.22/0.8655 | 36.03/0.9320 | 26.28/0.7888 | 38.70/0.9589 | 36.42/0.9323 | 26.54/0.8057 - |
| SkipVSR-BasicVSR | 0.93 | 264.0(78%) | 49 | 30.02/0.8608 | 35.90/0.9315 | 26.09/0.7742 | 38.46/0.9534 | 36.09/0.9279 | 26.23/0.7989 |
| SkipVSR-BasicVSR | 0.94 | 216.6(63%) | 42 | 29.76/0.8547 | 35.62/0.9248 | 25.67/0.7651 | 38.03/0.9487 | 35.68/0.9223 | 25.88/0.7890 |
| SkipVSR-BasicVSR | 0.95 | 145.5(43%) | 30 | 29.38/0.8468 | 35.02/0.9136 | 25.22/0.7450 | 37.58/0.9400 | 35.13/0.9117 | 25.57/0.7777 |

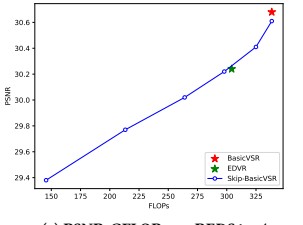

(a) PSNR-GFLOPs on REDS4 ×4

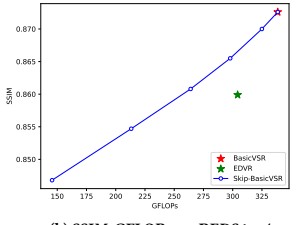

(b) SSIM-GFLOPs on REDS4 ×4

Figure 4: Quantitative results of performance-efficiency trade-off. We apply SkipVSR to BasicVSR with scaling factors ×4 on REDS4 [19] dataset. For a fair comparison, we adopt BasicVSR [2] and EDVR [24] as label. Average FLOPs of all $180 \times 320$ LR patches and PSNR/SSIM calculated on the full image are reported.

original VSR networks while maintaining the performance. More important, our proposal enables to meet different computational resource scenarios based on training once. For example, our model only requires 43% of the original model to obtain recovery results. The computational cost of BasicVSR is significantly reduced by our method, and the computational costs of head and tail stay the same. **Scalability Results** We show the performance-efficiency trade-off results in Fig. 4 to demonstrate the scalability of SkipvSR. By controlling the confidence threshold during inference, SkipVSR can achieve scalable performance-efficency trade-off. Therefore, we can deploy one SkipVSR network on platforms with different computational resources. For the device with low computational resource, we can decrease the confidence threshold to get lower performance and faster inferences speed.

**Visual Results** Figure 6 shows the qualitative comparison of our method against the original VSR networks and with different confidence threshold. As we can see, SkipVSR can achieve same or even better visual results compared with original VSR networks. Figure

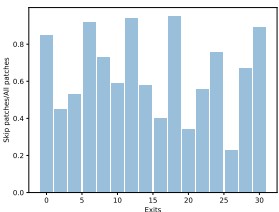
(a) Number of pass patches per layer

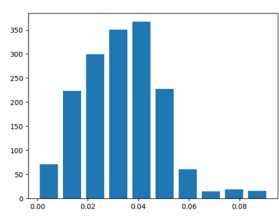
(b)The statistic of PSNR difference

Figure 5: (a) Statistical results of the number of pass patches per layer. We apply SkipVSR to BasicVSR with scaling factors ×4 on REDS4 dataset, and count the ratio of the number of patches executed per layer relative to the total patch numbers. (b) Statistical results of the intermediate feature difference bewteen layers distribution. We calculate the results intermediate feature difference $||H_i^S - H_{target}^O||_2^2$ between adjacent layers.

5-(b) visualizes the length of adaptive routing on each patch. As can be seen, the confidence selector network selects fewer blocks in the path for smooth regions. As for patches in complicated regions, the confidence selector network allocates more blocks. This is consistent with the motivation of applying appropriate networks to various difficulties. More important, we also visualize the number of patches executed by each block as figure 5-(a), and we can see that it is not the case that the later layers execute fewer blocks, which verifies the fact that we perform confidence per layer not exit once is correct.

## 4.3 Ablation Study

**The Validation of Components in SkipVSR.** We conduct an ablation study to demonstrate of our SkipVSR by progressively adding components. The result are shown in table 3. VSR$_1$ adopts skip blocks uniformly while VSR$_2$ adopts confidence estimator to guide which block should be skipped. Comparing VSR$_1$ and VSR$_2$, it is indicated that our strategy is superior to other skip strategy and provide a promising direction for VSR inference speed-up. That is because confidence estimator can extract all information contained in feature channels and analyze its restoration performance. As we can see, SSL$_3$ and SSL$_4$ outperform other methods. That is because introducing the temporal feature alignment can mitigate error accumulation owing to layer skipping. Comparing VSR$_3$ and VSR$_4$, we can see that With the guarantee of temporal feature alignment, adding a confidence estimation module does not affect the overall performance either.

**Variants of skip strategy** To validate our adaptive patch routing strategy, we conduct experiments on REDS4 for BasicVSR with different skip strategies. We compare our approach with three empirical policies inllustrated in Table 4: skip blocks uniformly, shallow and deep aggressively execute shallow and deep layers respectively. It shows that using our strategy can substantially reduce the computation cost while maintaining performance better. For a more fair comparison, we set the confidence upper limits with 0.95 and 0.94. It is indicated that while confidence threshold set as 0.94, SkipVSR

Table 3: Validation of the components in our SkipVSR. PSNR(dB) results evaluated on REDS4[19] (4×). The backbone is BasicVSR[2].

| Methods | VSR$_1$ | VSR$_2$ | VSR$_3$ | VSR$_4$($Ours$) |
|---|---|---|---|---|
| Uniform Skip | ✓ | | | |
| Confidence Estimator | | ✓ | | ✓ |
| Temporal Alignment | ✓ | | ✓ | ✓ |
| PSNR(dB) | 25.30 | 29.89 | **30.60** | **30.60** |

Table 4: Adaptive inference policy comparison of SkipVSR on REDS4. The index of threshold corresponds to the upper limits of confidence value in inference. Uniform, shallow and deep skip represents skip randomly, only execute the second half of the model and the first half of the model, respectively. For both shallow network and deep networks, SkipVSR outperforms these strategies by a large margin.

| Skip strategies | REDS4 | | |
|---|---|---|---|
| | FLOPs | Runtimes | PSNR/SSIM |
| uniform skip | 278G | 35ms | 25.30dB/0.7288 |
| shallow skip | 278G | 34ms | 12.01dB/0.1840 |
| deep skip | 278G | 35ms | 28.87dB/0.7930 |
| SkipVSR(threshold = 0.94) | 293G | 42ms | **29.76dB/0.8547** |
| SkipVSR(threshold = 0.95) | **270G** | **30ms** | 29.38dB/0.8468 |

Table 5: The ablation study of confidence estimator with different GTs. PNSR, PSNR interval, Feature, Feature interval represents the average difference of final PSNR value, PSNR value based on intermediate feature, final feature and intermediate feature, respectively.

| The GT of Confidence Estimator | REDS4 | | |
|---|---|---|---|
| | FLOPs | Runtimes | PSNR/SSIM |
| PSNR | 338.5G | 57ms | 25.67dB/0.6789 |
| PSNR Interval | 338.5G | 103ms | **30.62dB/0.8725** |
| Feature | 338.5G | 58ms | 21.01dB/0.3031 |
| Feature Interval | 338.5G | 58ms | 30.60dB/**0.8726** |

can achieve an PSNR index of more than about 0.5dB with less GFLOPs consumption as well as faster inference. We outperform other strategies by a large margin.

**Variants of Confidence Estimator** In order to improve the accuracy and accelerate the convergence of the confidence estimator, we apply different GT designs about feature and performance index (PSNR) to supervise the training of confidence estimator. So we perform ablation experiments on varaints GT supervision with confidence estimator. As shown in Table 5, we adopt the final PSNR difference between original model and SkipVSR as equation 14,

$$S_i = 1 - \theta||PSNR_N^S - PSNR_N^O||_2^2 \qquad (14)$$

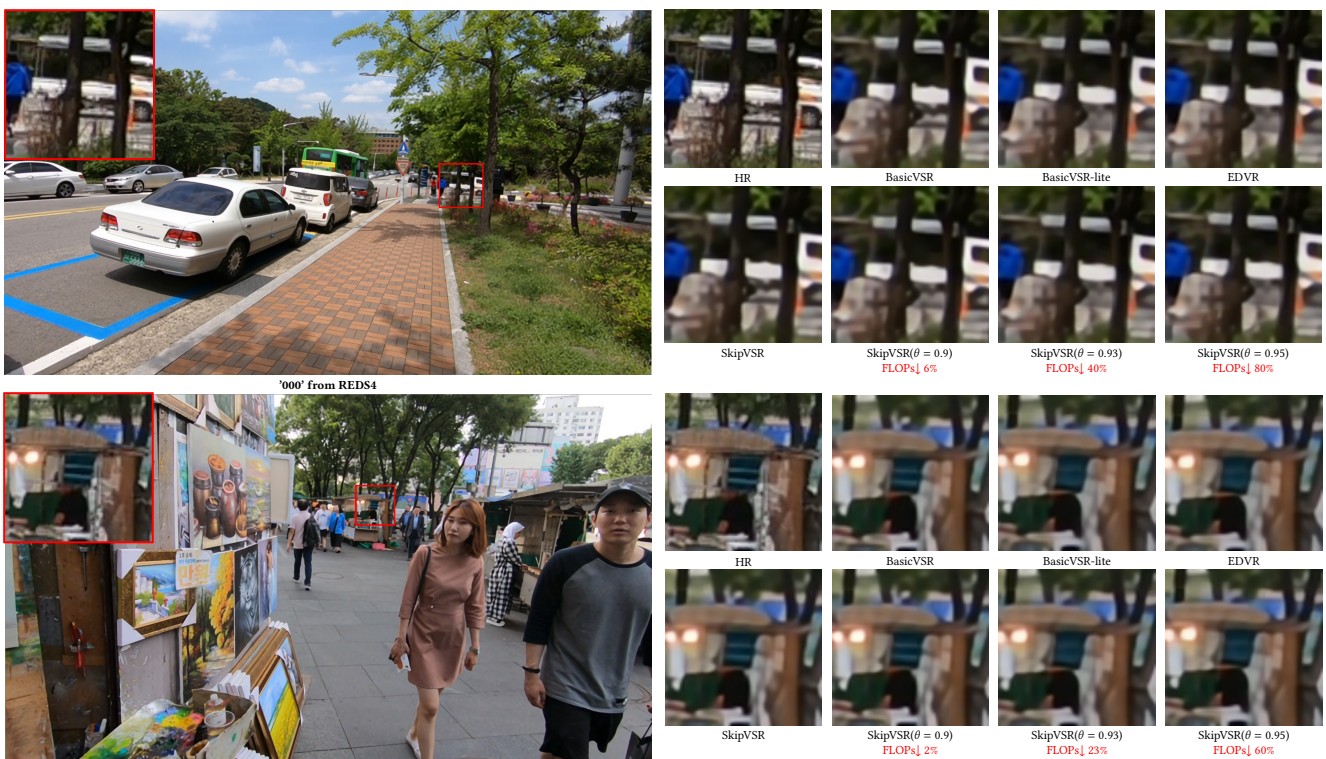

**Figure 6: Qualitative comparison between various VSR and our skipping strategy with different confidence threshold on REDS4 [19]. As we can see, our method maintains comparable visualization when the number of FLOPs is reduced by up to 80%.**

the intermediate PSNR value based on features extracted by intermediate layers as equation 15,

$$S_i = 1 - \theta||PSNR_i^S - PSNR_i^O||_2^2 \qquad (15)$$

the final feature difference between original model and SkipVSR as follow:

$$S_i = 1 - \theta||H_N^S - H_N^O||_2^2. \qquad (16)$$

and the intermediate feature difference as equation ??. As shown in table 5, it shows that only adapting the metric indexes calculated at the end of whole model can not provide great guidance the confidence estimators for each intermediate layer, causing a failure of the whole model to converge. The PSNR index obtained by upsampling the intermediate features can be a good instruction for model recovery, which achieve 30.62dB PSNR. However, the resource consumption caused by upsampling each layer is huge. As a trade-off, we use the intermediate features as confidence estimator GT, which give an enough and clear guidance efficiently. getting 30.60dB with 58ms.

### 4.4 Limitations

There are several limitations in our work. 1) The skipping operation only performs on the block-wise level. Future work should take more fine-grained skipping into consideration. 2) More lighter weight temporal feature alignment strategies can be explored, allowing more accurate alignment of features with less time and resource consumption. We believe that our research is inspiring and the above limitations should be addressed in future works.

**Table 6: The ablation study of temporal feature alignment is set per several layers. Feature intervals represent the number of layers to be constrained. '−' indicates that the model can not be converged.**

| Feature Intervals | REDS4 | | |
|---|---|---|---|
| | FLOPs | Runtimes | PSNR/SSIM |
| 15 | 338.5G | 58ms | – / – |
| 10 | 338.5G | 58ms | 25.52dB/0.6525 |
| 6 | 338.5G | 58ms | **30.60dB/0.8726** |

## 5 CONCLUSION

In this paper, we propose an adaptive patch routing strategy to speed up the inference of the VSR model in resource-limited situations. Specifically, the key idea is using a confidence estimator constrained by a feature difference loss between current layers and target layers to obtain the binary decision, which provides a execution signal for each patch in each images to indicate its recovery importance. What's more, temporal feature alignment is designed for keeping the accuracy and performance of target layers. We apply SkipVSR on the BasicVSR, and extensive experiments show that our proposal can achieve superior trade-off between performance and efficiency.

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
