# OpenReview forum: "SkipVSR: Adaptive Patch Routing for Video Super-Resolution with Inter-Frame Mask"
_acmmm.org/ACMMM/2024/Conference — MM2024 Poster_

### Official Review · Reviewer_H6Gg · 2024-05-12

**Rating:** 5
**Confidence:** 2

**Summary:**

This paper introduces a method to accelerate VSR networks by selectively bypassing certain neural network blocks, thereby reducing inter-frame redundancy. The framework efficiently determines the patch routing by leveraging feature similarity. To address the issue of error accumulation, the authors have introduced a temporal feature alignment strategy, which involves incorporating an additional loss term to minimize the distance between the dynamic network and the full networks. The findings of this study indicate that the framework significantly enhances the speed of the VSR process while maintaining satisfactory video quality.

**Strengths:**

1. The task (accelerating VSR network) is significant and the proposed approach is valid.
2. The design of the framework is intuitive and easy to follow
3. The high scalability of the framework is quite beneficial in the context of VSR acceleration.
4. Comprehensive evaluations have been done to demonstrate the effectiveness of the proposed approach

**Limitations:**

1. The proposed method has only been tested with BasicVSR, yet the framework appears to have broader applicability. It would be helpful if the authors could elaborate on how SkipVSR might be adapted to work with other VSR networks.

2. Inter-frame dynamics are a critical factor in video super-resolution. Could the authors detail how well SkipVSR handles videos that exhibit varying levels of temporal dynamics? This area seems to lack sufficient experimentation and discussion in the current study.

3. Typos and formatting errors:
(1) L24: "Temporal" miscapitalized
(2) PSNR: please provide the full name of this term
(3) L511: wrong references
(4) Table 1: numeric changes are not marked in red or blue as noted in the caption.

**Suitability:**

3

---

### Official Review · Reviewer_4HTq · 2024-05-23

**Rating:** 3
**Confidence:** 3

**Summary:**

The paper presents SkipVSR, a novel approach for accelerating video super-resolution (VSR) through adaptive patch routing. By dynamically skipping certain blocks during inference based on a confidence estimator, SkipVSR aims to reduce computational cost while maintaining high reconstruction accuracy. The method incorporates temporal feature distillation to address error amplification in recurrent networks. The proposed approach is evaluated using the BasicVSR model and demonstrates a 20% average speedup with comparable performance on various datasets.

**Strengths:**

- **Well-Structured and Clear:** The paper is well-structured and clearly written.

- **Computational Efficiency**: The proposed method significantly reduces computational costs, making it feasible for real-time applications

- **Honest Discussion of Limitations**: The authors provide a candid discussion of the method's limitations, demonstrating a balanced perspective.

**Limitations:**

- **Insufficient evaluation:** There is a lack of comparisons with recent relevant works beyond 2022, such as those in references [1-3].

- **Limited Generality**: The method is evaluated primarily on BasicVSR; its generalizability to other VSR models [4] or different tasks is not thoroughly explored.

- **Lack of related works**: The paper introduce a decision mechanism after each residual block to determine whether next residual block should be executed or skipped. It would benefit from a discussion on adaptive or dynamic inference, referencing works like [5-6].



[1] Structured sparsity learning for efficient video super-resolution. Proceedings of the IEEE/CVF Conference on Computer Vision and Pattern Recognition. 2023.

[2] Multi-scale video super-resolution transformer with polynomial approximation. IEEE Transactions on Circuits and Systems for Video Technology (2023).

[3] Multi-frequency representation enhancement with privilege information for video super-resolution. Proceedings of the IEEE/CVF International Conference on Computer Vision. 2023.

[4] Compression-aware video super-resolution. Proceedings of the IEEE/CVF Conference on Computer Vision and Pattern Recognition. 2023.

[5] Lgvit: Dynamic early exiting for accelerating vision transformer. Proceedings of the 31st ACM International Conference on Multimedia. 2023.

[6] SAVSR: Arbitrary-Scale Video Super-Resolution via a Learned Scale-Adaptive Network. Proceedings of the AAAI Conference on Artificial Intelligence. Vol. 38. No. 4. 2024.



### Minor points

- In Line 13, "which drag down" should be "...drags..."
- In Line 130, "...exexuted..." should be "...executed..."
- In Line 732, "The result are shown in table 3" should be "The results are shown in Table 3"

- In Line 511, "According to previous methods EDVR [? ] and BasicVSR [? ],", which lacks reference labels.

**Suitability:**

2

---

### Official Review · Reviewer_kRK9 · 2024-05-25

**Rating:** 4
**Confidence:** 3

**Summary:**

The paper introduces an adaptive and efficient acceleration strategy called SkipVSR, which is based on the BasicVSR model. It is claimed to be the first method to design an adaptive acceleration strategy for video super-resolution (VSR) tasks. The paper describes the novel aspects of its patch-predicting algorithm, which includes a confidence estimator module to predict the importance of each patch in a video frame, and a feature alignment module to correct errors and maintain accuracy when skipping blocks in a recurrent VSR network. The experiments demonstrate that SkipVSR can significantly reduce the computational cost of VSR models while maintaining performance. Additionally, SkipVSR is scalable and adjustable, allowing it to be configured for different speed-performance trade-offs based on various computational budgets and confidence thresholds.

**Strengths:**

1. The paper provides a strong motivation for reducing the computational cost of VSR models. While VSR can be highly beneficial for mobile devices, previous models can cause significant delays and power consumption.

2. The adjustable confidence threshold offers configuration options suitable for different devices, enhancing the practicality of the proposed method.

3. It appears that implementing this method on previous VSR models with similar architectures would be relatively easy， as it involves additional modules only.

4. The temporal feature alignment module utilizes the original network with superior feature representations to constrain the lighter adaptive network, which demonstrates a general concept of knowledge distillation.

**Limitations:**

1. The trade-off presented in Table 2 shows a noticeable decline in performance. When the inference speed is doubled, the PSNR drops by about 1.5dB across each dataset. Under certain configurations, the effectiveness of the method is not convincing. For example, the RRN method in Table 1 achieves a PSNR of 38.96 with a runtime of 45ms, whereas the threshold 0.93 SkipVSR has a similar runtime of 49ms but only achieves a PSNR of 38.46 on the UDM10 dataset. The performance gap is considerable and should be explained by the authors.

2. Moreover, since SkipVSR is based on BasicVSR, it is important to comprehensively compare the customized model with the base model to prove the effectiveness of the customization. For instance, one could reduce the number of layers in the BasicVSR model to achieve a similar runtime as different SkipVSR configurations and then compare their performance. The existing results only show a slight performance decline without skipping operations (threshold=0), which is due to the added modules. Therefore, the experiment is insufficient.

3. As mentioned in the paper, the content of frames can significantly influence the frame-selecting module. Conducting experiments on different video content types (such as sports, films, etc.) would be more convincing than using standard datasets alone. The performance of this framework may vary across different video categories.

4. From the previous experiments, it appears that SkipVSR is capable of adapting to different trade-offs between performance and speed through simple configuration. However, the performance of the framework is not necessarily superior to the BasicVSR model.

5. Several mistakes should be addressed. Table 1 does not contain elements marked in red or blue as described.

**Suitability:**

2

---

### Official Review · Reviewer_xm3E · 2024-05-27

**Rating:** 1
**Confidence:** 2

**Summary:**

The paper introduces SkipVSR, a method that speeds up video super resolution by deciding which parts of the video data need more processing and which do not. This approach helps save computing resources without losing much quality in the video output, especialy for devices with limited processing power, like smartphones and tablets. The method builds on an existing model called BasicVSR and modifies it to only work on important parts of the video, making the processing faster and efficient. Experiments suggests that SkipVSR can achieve 20% average speedup of the super-resolution process comparing to the baseline.

**Strengths:**

1. This paper proposes using a confidence estimator to dynamically manage the execution of network blocks during the super-resolution process. This approach effectively balances performance with computational cost.

2. The paper assesses the method against several benchmarks, including REDS4 and Vimeo-90K, demonstrating its effectiveness and robustness across various scenarios and datasets. The performance results indicate that SkipVSR can achieve significant efficiency gains.

**Limitations:**

1. The contributions of this work are somewhat limited, mainly offering incremental improvements on the existing BasicVSR framework, which has already established the fundamental structure for video super-resolution.

2. The presentation of the paper is unclear. The notation used is inconsistent, for example, $h^f_t$ denotes the frame number using subscripts in section 3.1, whereas $h^t_m$ uses superscripts for the same purpose in section 3.2. This inconsistent notation makes the logic in section 3 hard to follow. Additionally, the descriptions and labels in Figure 3 do not match the text, with missing parts (a) and (b) that are referenced in the caption. It is also unclear where the confidence estimator fits within the diagram shown in Figure 3. It would be better if the authors can clarify the main logic of Section 3 with consistent notations.

**Suitability:**

3

---

### Meta-Review · Area_Chair_PDMC · 2024-07-01

**Recommendation:** Accept (Poster)
**Confidence:** 4

**Metareview:**

This paper presents SkipVSR, an adaptive acceleration strategy for video super-resolution that aims to reduce computational costs while maintaining performance. The method demonstrates promising results with a 20% average speedup compared to the baseline, and offers scalability for different speed-performance trade-offs. However, reviewers noted concerns about limited generalizability beyond BasicVSR, insufficient comparisons with recent works. Despite these limitations, the paper's innovative approach to VSR acceleration and its potential practical applications make it an accept.

Additional comment form SAC: Despite its strengths, the submission lacks multi-modal, multi-media aspects; its focus on "video super-resolution" is inherently single media. Therefore, we recommended that its is accepted as a "Poster"